# Mechanical Performance Study of Beam–Column Connection with U-Shaped Steel Damper

**DOI:** 10.3390/ma15207085

**Published:** 2022-10-12

**Authors:** Chun-Xu Qu, Yu-Wen Xu, Jin-He Gao, Wei-Hao Zhou, Bao-Zhu Zheng, Peng Li

**Affiliations:** 1School of Civil Engineering, Dalian University of Technology, Dalian 116023, China; 2School of Civil and Architectural Engineering, East China University of Technology, Nanchang 344000, China

**Keywords:** U-shaped steel damper, quasi-static testing, mechanical property, semi-rigid connection, energy dissipation

## Abstract

The article proposes the use of a semi-rigid energy-dissipation connection combined with a U-shaped metal damper to avoid brittle failure of rigid steel beam–column connections under seismic loading. The U-shaped metal damper connects the H-section column and the H-section beam to form a new energy-dissipation connection as an energy-dissipation member. Compared with the existing research, this connection has a stable energy-dissipation performance and great ductility. To clarify the mechanism of energy dissipation, mechanical models under two U-shaped damping deformation modes are established. The calculation formulas for the yield load and stiffness are derived for the corresponding deformation mode using the unit load method. Taking the T-shaped beam–column connection and the application of U-shaped steel damper in the beam–column connection as an example, the mechanical model of the connection is established and the calculation formulas for the yield load and stiffness are derived. At the same time, the connection is subjected to a quasi-static test under cyclic loading. The results show that the hysteretic curve of the test is complete and that the skeleton curve is accurate compared to the theory. The error range of the initial stiffness and yield load obtained by the test and the theoretical formula is kept within 20%, indicating that the theoretical formula is reasonable and feasible. In addition, the correctness of the finite element model is verified by establishing a finite element model and comparing it with the test. The mechanical responses of purely rigid connections and rigid semi-rigid composite connections are compared and analyzed using a multi-story and multi-span plane frame as an example. The results show that the model with semi-rigid connections, compared to the model with rigid connections, avoids the gradual loss of bearing capacity caused by the failure of the connection area of the second floor of the main structure and improves the seismic performance of the main structure.

## 1. Introduction

Earthquakes are a common natural phenomenon and pose a significant threat to the structural safety of buildings [1,2,3,4,5,6]. One of the most important structural elements in a steel frame building is the beam–column connection. Most rigid connections in traditional connection construction are hybrid stud-welded connections or pure welded connections. Due to the Northridge earthquake in the United States in 1994 and the Kobe earthquake in Japan in 1995, the beam–column connections of traditional steel frame buildings suffered brittle failure because they were unable to absorb the seismic energy by plastic deformation of the structural members [7,8]. Since then, much research has been conducted on beam–column connections in steel structures. The main concepts are as follows: improving the details of the welded hybrid bolted connection [9,10,11], strengthening the connection [12,13,14,15], or weakening the beam flange [16,17] by relocating the plastic hinge outward. On the other hand, repairing plastic damage to steel beams leads to destruction of building functions and high repair costs [18,19].

In recent years, a new connection system based on the concept of passive control has attracted the interest of scholars. In this connection system, the damper and the beam–column connection are combined. The good energy-dissipation performance of the damper can improve the energy-dissipation capacity of the connection, protect the main structure from damage, and enable rapid repair of the structural system after an earthquake. For example, Koetaka et al. designed a π-type damping connection for the weak axis connection of the beam column. They installed a π-type metal damper on the beam flange near the beam–column connection and investigated the mechanical properties of the damping connection using a cyclic loading test [20]. Qu et al. proposes a double-homotopy method to solve the BMI problem in discrete time domain [21]. Huo et al. presents a robust H∞ controller design for civil structures with consideration of the parametric uncertainties through the linear fractional transformations approach [22]. Huang et al. and Zhou et al. investigated the mechanical properties of energy-dissipation joints made of steel plates in wood frame construction by quasi-static tests [23,24]. Zhou et al. and Wu et al. conducted low cyclic loading tests on beam–column connections of fan-shaped viscoelastic damping concrete frames [25,26]. Chen et al. strengthened the purlin roof of a tiled wood structure by installing damping limiters in the overlap connection between the eaves and the wood purlins [27].

Metal dampers are a type of damper that are commonly used [28,29]. Due to their good energy-dissipation performance and economy, they are widely used in the seismic design of buildings. Wu et al. proposed connecting the T-shaped beam support to the floor plate, which improves the deformation capacity of the connection [30]; however, the deformation capacity of the steel plate connected to the web of this connection is very small, making it difficult to repair the connection. Hsu and Halim proposed a curved damper for frame connections to improve energy dissipation, but the fabrication is complex and not suitable for large deformations [31]. Saffari and Hedayat proposed a beam–column connection with a circular hole damper at the top and bottom beam-flange edges. Although it can protect the beam–column connection, it is difficult to repair and takes up a lot of space [32].

To solve the problems of brittle failure, poor deformation ability, and complex post-earthquake repair with traditional beam–column connections, this study proposes a new beam–column connection combined with a U-shaped steel damper, which can improve the seismic performance of the structure and facilitate repair after an earthquake. In this study, the mechanical properties of beam–column connections with U-shaped steel dampers are investigated by a combination of theoretical research, experiments, and numerical analysis. The mechanical responses of purely rigid connections and rigid semi-rigid composite connections are compared and analyzed using a multi-story and multi-span plane frame as an example.

## 2. U-Shaped Steel-Damper Model

The U-shaped steel damper in this beam–column connection of the steel structure is subjected to loads in both vertical and horizontal directions. Therefore, mechanical models with two deformation modes are created to account for its mechanical properties in both directions.

### 2.1. Vertical Direction

Figure 1 shows the vertical model of the U-shaped steel damper. The bending moments of the vertical load *P_v_* on the circular arc section and the straight section of the damper are *M_v_*_1_ and *M_v_*_2_, respectively. The Equations (1) and (2) are as follows:(1)$$M_{v1}=P_{v}r\left(1-\cos\varphi \right)$$
(2)$$M_{v2}=2P_{v}r$$
where *r* = radius.

According to the unit load method, the displacement *v* generated by the action point P under the vertical load *P_v_* is expressed by the following Equation (3):(3)$$v=\int_{0}^{\pi }\frac{P_{v}r^{2}{\left(1-\cos\varphi \right)}^{2}}{EI}rd\varphi +\int_{0}^{l}\frac{P_{v}{\left(2r\right)}^{2}}{EI}dx\;=\frac{9\pi r^{3}+8l^{3}}{6EI}P_{v}$$

Its vertical stiffness *K_v_* is expressed by the following Equation (4):(4)$$K_{v}=\frac{P_{v}}{v}=\frac{6EI}{9\pi r^{3}+8lr^{2}}$$
where *r* = radius;*l* = length of the straight section;*I* = moment of inertia of the section.

### 2.2. Horizontal Direction

Figure 2 shows the horizontal model of the U-shaped steel damper. The bending moments of the horizontal load *P_u_* on the circular-arc section and the straight section of the damper are *M_u_*_1_ and *M_u_*_2_, respectively. The Equations (5) and (6) are as follows:(5)$$M_{u1}=P_{u}\left(l+r\sin\varphi \right)$$
(6)$$M_{u2}=P_{u}x$$

The displacement u of the action point P under the action of the horizontal load *P_u_* is expressed by the following Equation (7):(7)$$u=2\int_{0}^{l}\frac{P_{u}x^{2}}{EI}dx+\int_{0}^{\pi }\frac{P_{u}{\left(l+r\sin\varphi \right)}^{2}}{EI}rd\varphi \;=\frac{\frac{2l^{3}}{3}+\pi rl^{2}+\frac{\pi r^{3}}{2}+4r^{2}l}{EI}$$

Its horizontal stiffness *K_u_* is expressed by the following Equation (8):(8)$$K_{u}=\frac{P_{u}}{u}=\frac{EI}{\frac{2l^{3}}{3}+\pi rl^{2}+\frac{\pi r^{3}}{2}+4r^{2}l}$$

## 3. Quasi-Static Testing

### 3.1. Test Equipment

The quasi-static test of the self-balancing reaction frame was completed (as shown in Figure 3). The servo actuator can apply a maximum pressure load of 1500 kN, a maximum tensile load of 700 kN, and a maximum stroke of 500 mm. The beam–column connection structure consists of an H-beam, an H-column, a U-shaped steel damper, and angle steel. Each element is held together by high-strength bolts, and the column base is held together by an in-plane hinge.

Figure 4 shows part of the test. Figure 5 shows the structure of the loading device and displacement meter in detail. To measure the horizontal displacement of the upper end of the beam, the D2 displacement meter is attached to the left side of the upper end of the beam. To measure the horizontal displacement of the lower end of the beam, the D2 displacement meter is placed on the right side of the lower end of the beam. The strain gauges are placed inside and outside the middle arc of the U-shaped damper to illustrate the mechanical response of the beam during loading.

The column cross-section is H350 × 350 × 12 × 19, as shown in Figure 5. The beam cross section is H350 × 175 × 7 × 11. The angle steel sections are L120 × 20 and L250 × 150 × 20. The length of the column is 1800 mm, and the length of the beam is 1500 mm. The thickness t of the U-shaped metal damper is 20 mm, as shown in Figure 6. The radius of the central axis of the arch part is 60 mm, the length L of the straight section is 100 mm, the radius of the opening of the straight section of the damper opening is 11 mm, and the distance l from the center of the damper opening to the junction of the straight section and the arch part is 80 mm. The parameters of the design are listed in Table 1.

Q345B is used for beams, columns, and angle steel. The yield strength of beams, columns, and angle steel is 384.31 N/mm^2,^ and the ultimate strength is 543.28 N/mm^2^, as shown in the material performance test. For high-strength bolts, 10.9S is used. The yield strength of high-strength bolts is 940 N/mm^2^, and the ultimate strength is 1040 N/mm^2^. The material of the U-shaped metal damper is Q235B, with modulus of elasticity 20565 Mpa, yield strength 306 N/mm^2^, and ultimate strength 471 N/mm^2^, as shown in Table 2.

### 3.2. Initial Stiffness

The mechanical properties of the connections between the beam and column under load are investigated. The deformation diagram is shown in Figure 7. If point a is the center of rotation and triangle abc is the rigid rotation, the horizontal displacement component of the displacement vector of point c should be set as *u* and the vertical displacement component as *v*.

The relationship between the horizontal displacement *u* and the vertical displacement *v* is shown in Figure 8. The Equation (9) is as follows:(9)$$u:v=d_{bc}:d_{ab}$$
where *d_ab_* = distance between the upper and lower flanges of the beam;

*d_bc_* = e center of the distance from the bolt center to the beam column.

**Figure 8 materials-15-07085-f008:**
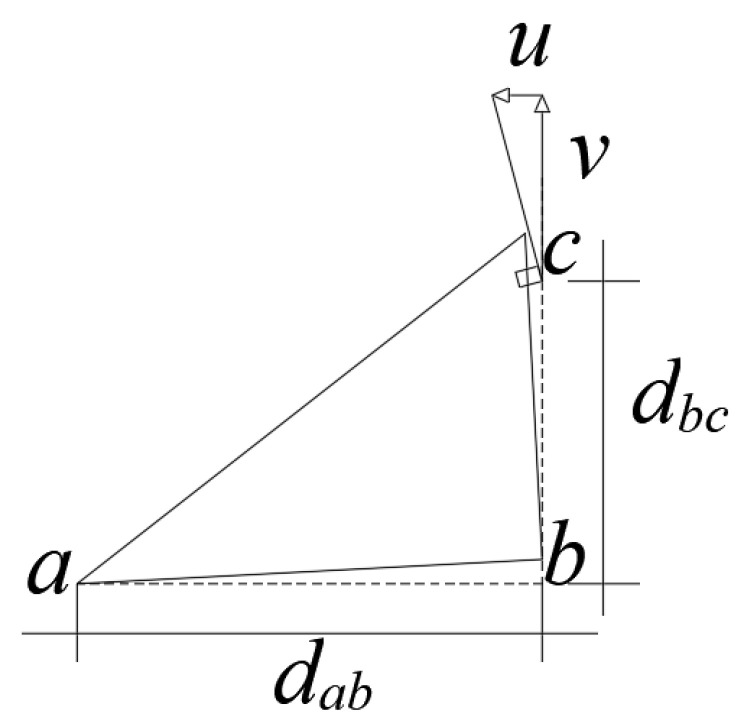
Rotational deformation.

The vertical distance from the stress point at the beam end to the rotation center is *h*. At point *c*, the horizontal load on the beam is *P_u_*, and the vertical load is *P_v_*. When the load applied by the hydraulic actuator to the beam end is *P*, the following Equation (10) can be obtained from the moment balance of the beam:(10)$$PH=P_{u}d_{bc}+P_{v}d_{ab}$$

Combine Equations (4) and (8)–(10) to obtain Equation (11):(11)$$u=\frac{PH}{K_{u}d_{bc}+K_{v}\frac{d_{ab}^{2}}{d_{bc}}}$$

The interlayer deformation angle is expressed by the following Equation (12):(12)$$\theta =\frac{u}{d_{bc}}\;$$

The initial stiffness Equation (13) can be obtained by combining Equations (11) and (12):(13)$$K=\frac{P}{\theta }=\frac{K_{u}d_{bc}^{2}+K_{v}d_{ab}^{2}}{H}$$

### 3.3. Yield Load

Based on the distribution law of bending moment and internal force, the bending moment of the circular arc-shaped section is the sum of Equations (1) and (5) and that of the straight section is the sum of Equations (2) and (6). The load is the yield load of the connection when the maximum bending moment reaches the full plastic bending moment of the U-shaped steel damper. The full plastic bending moment *Mp* is determined by the yield criterion for the entire section. It can be defined by Equation (14):(14)$$M_{p}=\frac{f_{y}t^{2}l}{4}$$
where the yield strength is 306 Mpa, according to the material properties.

The bending moment ms of the arc section can be obtained from the following Equation (15):(15)$$M_{s}=P_{v}r(1-\cos\varphi )-P_{u}(l+r\sin\varphi )\;=\left(2rK_{v}\frac{d_{ab}}{d_{bc}}-K_{u}l\right)u$$

The bending moment mL of the straight section can be obtained from the following Equation (16):(16)$$M_{l}=P_{u}x-2P_{v}r\;=P_{u}r-2P_{v}r\;=(K_{u}r-2rK_{v}\frac{d_{ab}}{d_{bc}})u$$when *M* = *MP*, the U-shaped damper yields, and the horizontal yield displacement *U* of the damper can be obtained. At the same time, the yield displacement *U* of the beam section can be obtained from the following Equation (17):(17)$$U=u\frac{H}{d_{bc}}$$

The yield load expression is as follows Equation (18):(18)$$F=\frac{KU}{H}$$

### 3.4. Test Loading System

The horizontal reciprocating load *P* is applied to the top of the column. Displacement control is used in the loading system. The horizontal displacement is controlled by the actuator of the electro-hydraulic servo control system. Figure 9 depicts the loading system. The interlaminar deformation angle R (=δ/1500) reaches ±0.005 rad, ±0.01 rad, ±0.015 rad, ±0.02 rad, and ±0.05 rad at each level and loads once.

## 4. Mechanical Property Analysis

### 4.1. Result Analysis

Figure 10 depicts the test hysteresis curve. The figure shows that the hysteretic curve of the metal damping semi-rigid connection specimen has a spindle shape and no obvious slip phenomenon. It has excellent energy-dissipation and shock-absorption properties. The semi-rigid connection formed by installing the U-shaped metal damper in the steel structure’s beam–column connection has the effect of energy dissipation, and the U-shaped metal damper can absorb seismic energy.

Figure 11 depicts the test and theory skeleton curves. Table 3 shows that the error between the initial stiffness of the test and the theory is 4.45%, and the error between the yield load of the test and the theory is 18.21%, both of which are kept within 20%, demonstrating the theory’s applicability.

### 4.2. Seismic Performance Parameter Analysis

The equivalent viscous damping coefficient and the ductility coefficient of the sample can be obtained by analyzing the hysteretic curve to further analyze and compare the energy-dissipation performance of the sample.

The equivalent viscous damping coefficient is an important index for determining the energy-dissipation capacity of earthquake-resistant structures or members. It can be defined by Equation (19):(19)$$\xi_{e}=\frac{E_{DS}}{4\pi E_{S}}$$
where *E_DS_* is the hysteretic damping energy-dissipation equal to the area surrounded by the curve at maximum displacement, and *E_S_* is the area surrounded by the secant stiffness at maximum displacement.

Because the equivalent viscous damping coefficient is proportional to displacement under cyclic loading, the greater the equivalent viscous damping coefficient, the better the energy-dissipation performance. Table 4 displays the results. The table shows that as the interlayer deformation angle *θ* increases, so does the equivalent viscous damping coefficient of the connection specimen as a whole. The equivalent viscous damping coefficient of the connection specimen increases significantly when the interlayer deformation angle *θ* is 0.02 rad, which is due to the energy-dissipation effect of the U-shaped damper in the plastic stage before the beam column members.

The ductility coefficient is an index used to assess a structure’s or member’s plastic deformation capacity under the action of bearing capacity. It can be defined by Equation (20):(20)$$\mu =\frac{X_{u}}{X_{y}}$$
where *X_u_* is the ultimate displacement of the specimen, and *X_y_* is the yield displacement of the specimen.

Table 5 displays the results of the calculation of the connection ductility coefficient. As shown in the table, the ductility coefficient of the connection is 2.6. The ductility coefficient of the damper structural design is less than four due to its small size. The ductility of the connection can be further improved by changing the size of the damper’s structural design.

## 5. Finite Element Analysis

### 5.1. Finite Element Model

Finite element analysis based on the ANSYS 2020 R1 software was used to verify the accuracy of the test. The numerical finite element analysis model is established based on the test and the material properties of steel and are defined according to the Table 2. The finite element loading system is consistent with the test. The elastic–plastic four-node SHELL181 element simulates the beam, column, angle steel, and U-shaped metal damper, while the beam element BEAM188 simulates the high-strength bolt. When geometric and material nonlinearity are considered, the material model uses the bilinear isotropic strengthening model with the von Mises yield criterion and a Poisson ratio of 0.3. The materials’ constitutive model is shown in Figure 12. The σy and εy are the yield stress and yield strain, and the σs and εs are the ultimate stress and strain. The εs of the steel is 0.02, and the εs of the high-strength bolts is 0.05 [33]. The strengthening modulus of the steel is 0.02E, and that of the high-strength bolts is 0.009E, wherein E is the elastic modulus, and the strengthening modulus represents the slope of the line segment in the plastic stage of the constitutive model. The horizontal reciprocating load is applied to the right side of the top of the column. Displacement control is used in the loading system. The loading system is consistent with Figure 9.

The shape of the element is controlled to be quadrilateral, taking into account the influence of calculation time and accuracy. The suitable mesh size is selected based on sensitivity analysis. The range of 6–14 mm is selected for the sensitivity analysis of the U-shaped steel damper, and the mesh size of 10 mm is closer to the test. The sensitivity analysis of beam column components is carried out within the range of 15–40 mm, and compared with the test, the mesh size of 25 mm is closer. Therefore, the U-shaped metal damper is densely mapped and meshed, with its element size limited to 10 mm. The beam and column components are subdivided into coarse mapping grids, with element sizes limited to 25 mm. Figure 13 depicts the finite element model. Furthermore, angle steel connects the beam column flange away from the U-shaped metal damper. The beam flange’s in-plane constraints were released in the finite element model.

The contact problem is also the interaction between two objects. The surface of one object is called the contact surface, and the surface of the other object is called the target surface, which constitutes a contact pair. Nonlinear contacts were set at the points of contact between the model parts to account for friction between the surfaces. The contacts are implemented using the TARGET170 and CONTA174 contact elements, which allow defining mutual contact pairs between two surfaces (face-to-face) and between a surface and an edge (edge-to-face). All contacts were set as frictional with a coefficient of friction value of 0.3.

In the finite element model, the degrees of freedom of the Shell181 element include three translational degrees of freedom and three rotational degrees of freedom. All degrees of freedom of the column were constrained at the lower flange, and only the in-plane constraint of the beam flange was released.

### 5.2. Result Analysis

Figure 14 depicts the finite element model’s stress distribution. In the arc section of the U-shaped metal damper, there is a high-stress response area. The stress response of the beam column members and angle steel is elastic, as predicted by the theoretical analysis. It is also confirmed that when U-shaped metal damping is installed at beam–column connections, it can absorb seismic energy through tension and compression yield deformation. Figure 15 depicts a comparison of the theory, test, and finite element skeleton curves. The figure shows that the finite element analysis results match the test results well; the initial stiffness inaccuracy is 12.7%, and the yield load inaccuracy is 17.3%. This demonstrates that the finite element model is accurate.

### 5.3. Mechanical Property Analysis of Plane Frame

On the basis of verifying the rationality of the beam–column connection model, using the multi-story and multi-span plane steel frame as the research object, the mechanical responses of rigid connections and rigid semi-rigid composite connections are compared by pushover analysis in the field of overall structural systems. The contact conditions, grid division, and element selection of the frame model are basically consistent with those of the connection.

#### 5.3.1. Frame Structure Design

The research object is the three-story, three-span steel frame structure depicted in Figure 16. The beam length is 6000 mm, and the column height is 4000 mm. H-350 × 175 × 7 × 11 is used in the beam section. For the column section, H-250 × 250 × 9 × 14 is used. The angle steel sections are L120 × 15 and L200 × 125 × 15. The column base is permanently attached. The rigid connection is used in model L-1. The beam flange farthest away from the U-shaped metal damper is connected with the column flange by angle steel to form a semi-rigid connection at the beam–column connection of model L-2 installed with the U-shaped metal damper. Furthermore, model L-2 employs a rigid semi-rigid composite connection combination mode, which means that only one middle span is designed as a semi-rigid connection, while the others are rigid connections.

#### 5.3.2. Finite Element Model of Frame Structure

The beam element BEAM88 is used for high-strength bolts, and the elastic–plastic four-node shell element SHELL81 is used for beam column members, angle steel, and U-shaped metal dampers. For grid division, mapping division is used. Fine division is used near the U-shaped metal damper and connection area, and coarse division is used in other areas. The finite element model is shown in Figure 15. For beam column and angle steel, Q235B is chosen. Because the U-shaped metal damper enters the yield stage before the beam column components, LY225 is chosen as the material, and the beam column, angle steel, and U-shaped metal damper are all elastic–plastic. The bilinear isotropic strengthening model obeying the Voin-Mie yield criterion is chosen as the material model’s nonlinearity in the finite element model, and both material and geometric nonlinearity are considered.

In the finite element model, the column-base section nodes use a fixed connection to limit all degrees of freedom. Out-of-plane constraints are imposed on all nodes of the finite element model to prevent the entire structure from deforming out of plane.

#### 5.3.3. Pushover Analysis

The monotonic loading mode is used in the finite element model of a multi-story and multi-span steel frame structure. According to the relevant building design codes, the ratio of the horizontal load values of the first, second, and third floors is 1:1.45:2.39, and the loading position is depicted in Figure 16a.

(1)The relationship curve between the story’s sheer force and the story’s angle of deformation is shown.

The response results of the horizontal load curve and the inter-story deformation angle curve of the multi-story and multi-frame structure in pushover analysis are shown in Table 6. Figure 17 depicts the load-displacement curves of model L-1 and L-2. The semi-rigid connection model L-2 has a lower initial stiffness and yield load than the rigid connection model L-1.

(2)Stress-response analysis

Table 6 shows that the model L-1 with rigid connections has entered the yield state when the interlaminar deformation angle is 0.0078 rad. The stress nephograms of the rigid connection model L-1 and the semi-rigid connection model L-2 are further analyzed to compare and analyze the difference in stress response and plastic failure path of the main structure when the U-shaped metal damper is installed (as shown in Figure 18). The rigid connection model L-1’s maximum stress value is located in the beam–column connection area on the left side of the first-floor span (red circle position), whereas the model L-2’s maximum stress value is located on the arc segment of the U-shaped metal damper on the right side of the second-floor span and the left side of the third-floor span (red circle position).

It can be seen that installing the U-shaped metal damper at the connection improves the seismic performance of the main structure. At the same time, according to Figure 18 and the subsequent overall structural stress response with increasing deformation, the plastic failure path of the rigid connection model L-1 begins at the mid-span connection region of the first floor and ends at the mid-span connection region of the second floor; the plastic failure path of the semi-rigid connection model L-2 installed with dampers begins at the U-shaped metal damper on the first floor and ends at the mid-span connection. The comprehensive comparative analysis demonstrates that the semi-rigid connection model prevents the gradual loss of bearing capacity caused by damage to the main structure’s first floor connection domain and improves the seismic performance of the main structure.

## 6. Conclusions

The mechanical properties of the beam–column connection combined with the U-shaped steel damper are studied using the inverted T-beam–column connection as the research object. The theoretical analysis is used to derive the formulas for initial stiffness and yield load, and the quasi-static test under cyclic load is compared to the theory. Simultaneously, a multi-story and multi-span frame model is created using finite element simulation to compare and analyze the mechanical response of the frame structure’s semi-rigid connections. The following are the main conclusions:(1)The theoretical formulas of relevant parameters of connections are derived using the unit load method, and the quasi-static test under cyclic load is performed. The error range of the initial stiffness and yield load obtained by the test and the theoretical formula is kept within 20%, indicating that the theoretical formula is reasonable and feasible.(2)The beam–column connection has full and stable hysteretic curves, good hysteretic properties, and good ductility in the cyclic loading test. The metal damper in the shape of a U can absorb energy before the beam and column yield, and the beam and column are always in elastic response.(3)According to the finite element analysis, the overall trend of the skeleton curve of the test is basically similar to that of the finite element calculation curve, and the peak point is also relatively consistent. The error range between the initial stiffness and yield load obtained from the test and the finite element calculation were kept within 20%. Thus, the correctness of the model is verified.(4)A numerical simulation of a multi-story and multi-span plane frame is performed on the basis of verifying the model’s correctness. In comparison to the rigid connection model, the U-shaped damper enters the yield energy dissipation before the main structure, preventing the main structure from gradually losing bearing capacity due to the failure of the first-floor connection domain and improving the seismic performance of the main structure. As a result, the main structure is safe from harm.

Based on the above analysis, it can be determined that the beam–column connection has stable energy-dissipation performance and good ductility performance. In later research, the plastic performance response law of the beam–column connection under different seismic-wave loads with different damping parameters can be analyzed.

## Figures and Tables

**Figure 1 materials-15-07085-f001:**
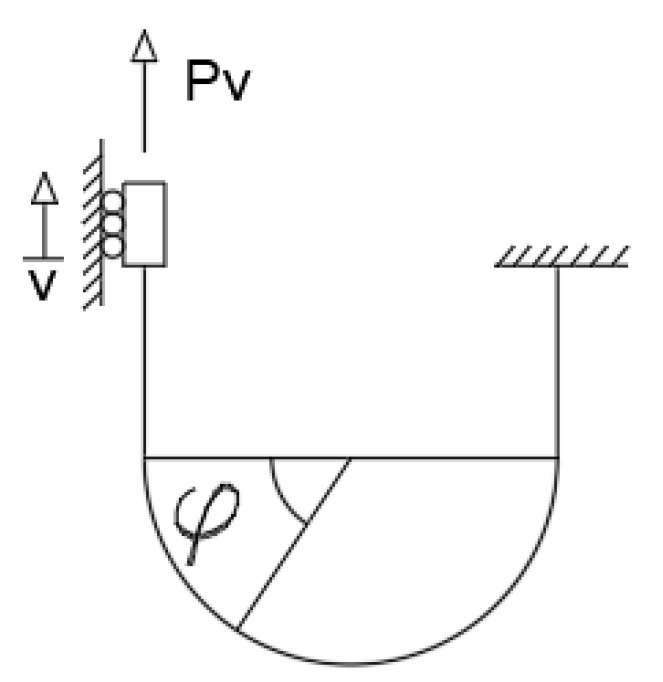
Vertical direction.

**Figure 2 materials-15-07085-f002:**
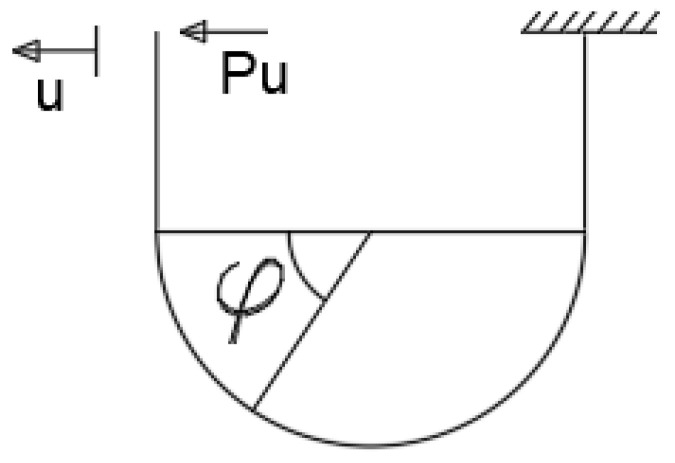
Horizontal direction.

**Figure 3 materials-15-07085-f003:**
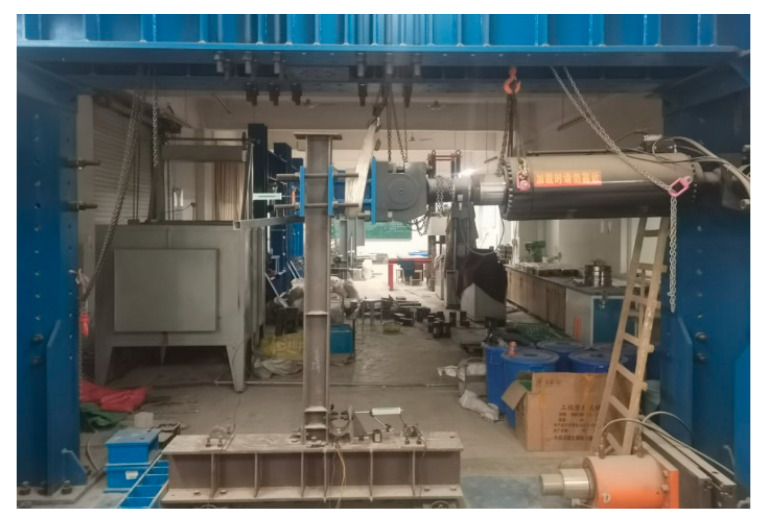
Test equipment.

**Figure 4 materials-15-07085-f004:**
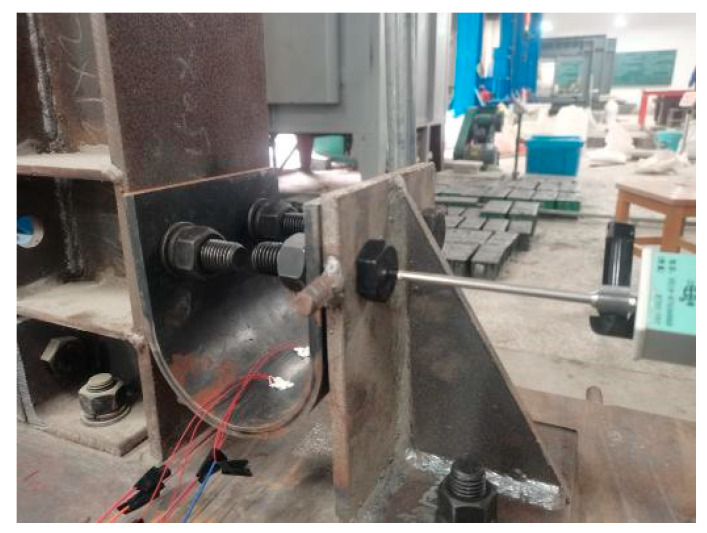
Detailed drawing.

**Figure 5 materials-15-07085-f005:**
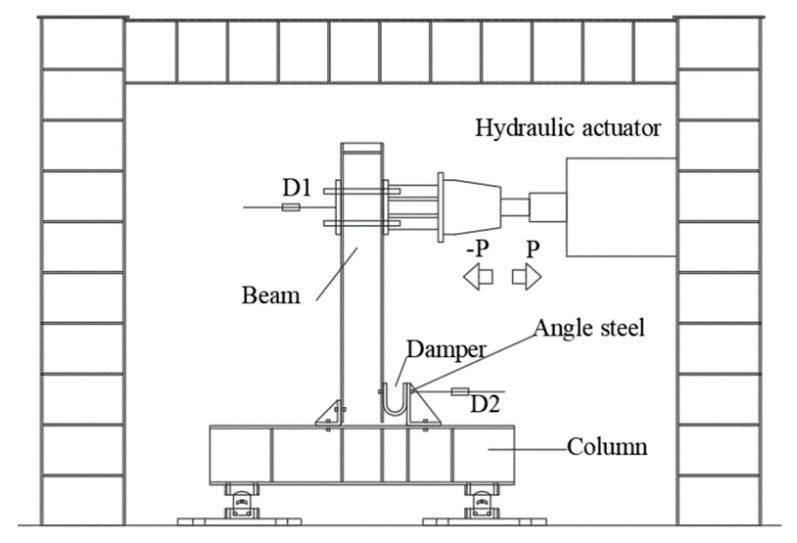
Position of the displacement meter.

**Figure 6 materials-15-07085-f006:**
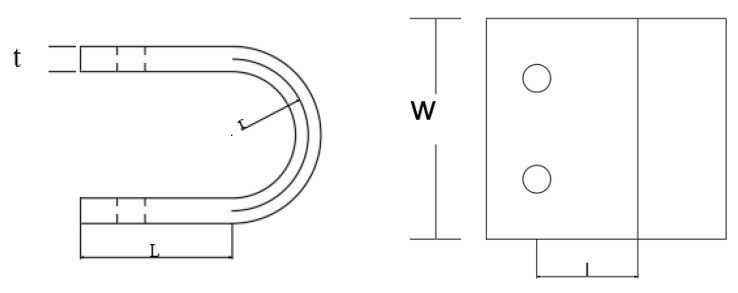
Damper construction.

**Figure 7 materials-15-07085-f007:**
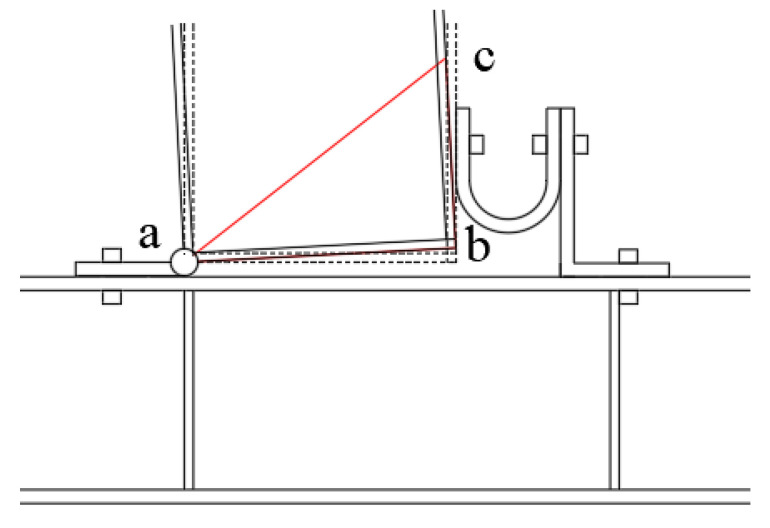
Connection deformation.

**Figure 9 materials-15-07085-f009:**
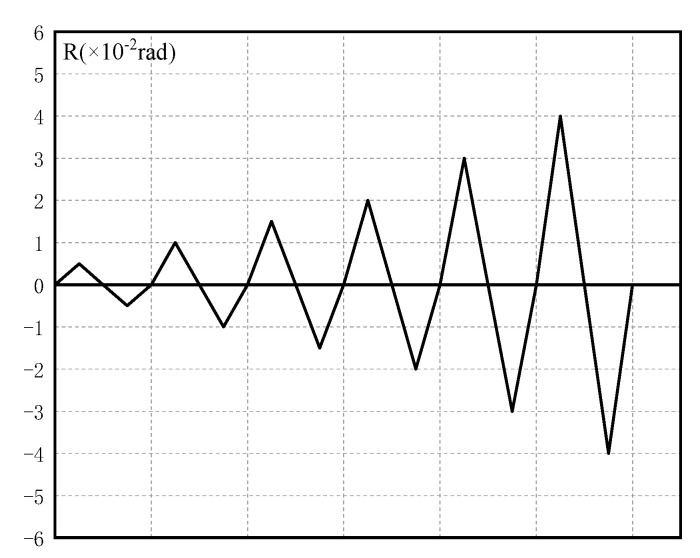
Loading protocol.

**Figure 10 materials-15-07085-f010:**
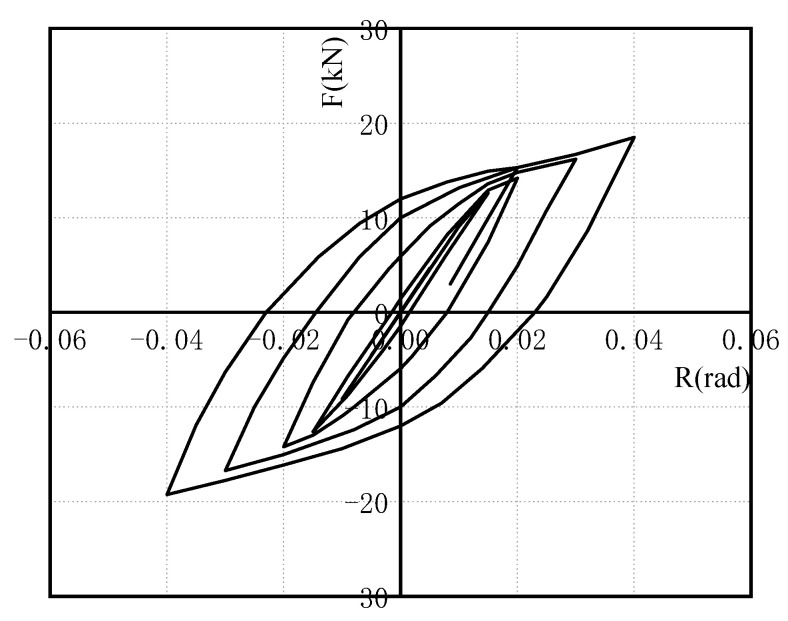
Hysteretic curve.

**Figure 11 materials-15-07085-f011:**
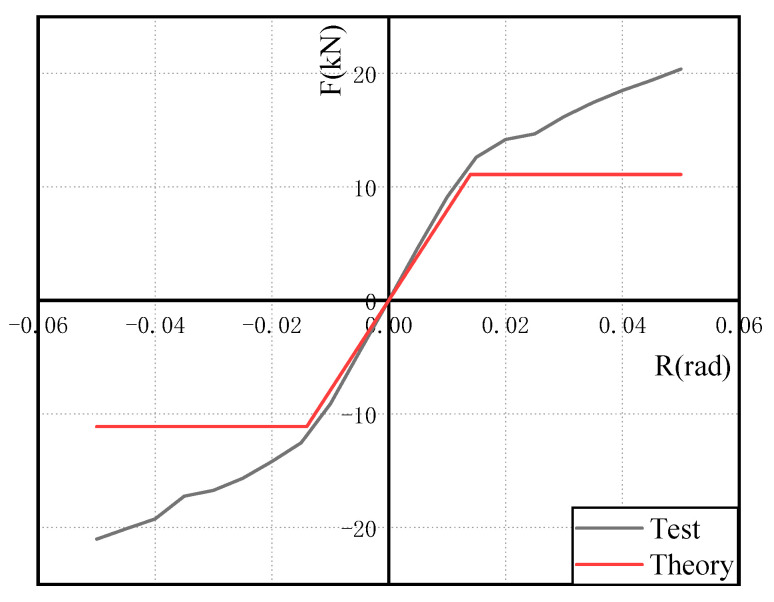
Skeleton curve.

**Figure 12 materials-15-07085-f012:**
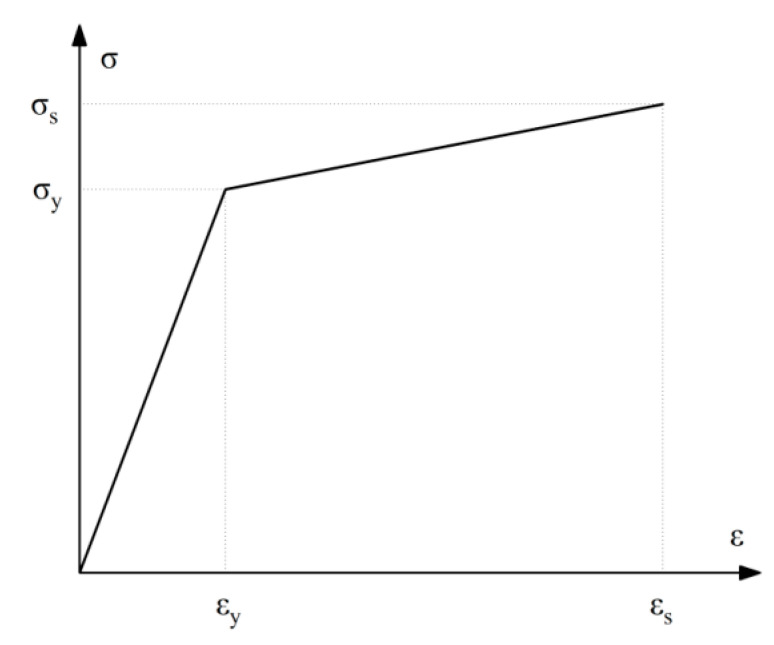
Materials’ constitutive model.

**Figure 13 materials-15-07085-f013:**
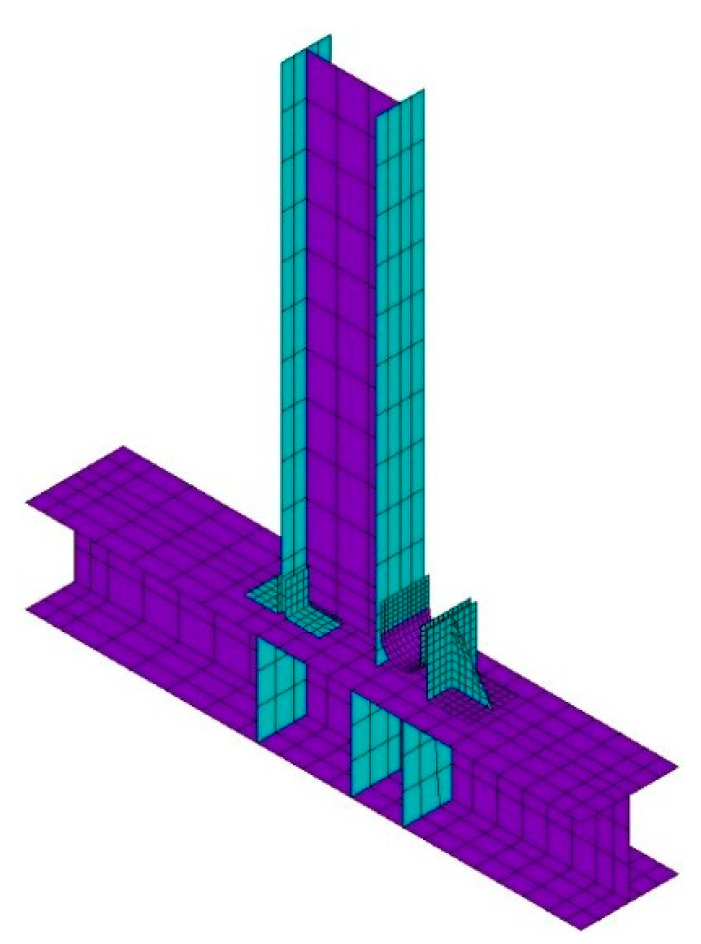
Finite element model.

**Figure 14 materials-15-07085-f014:**
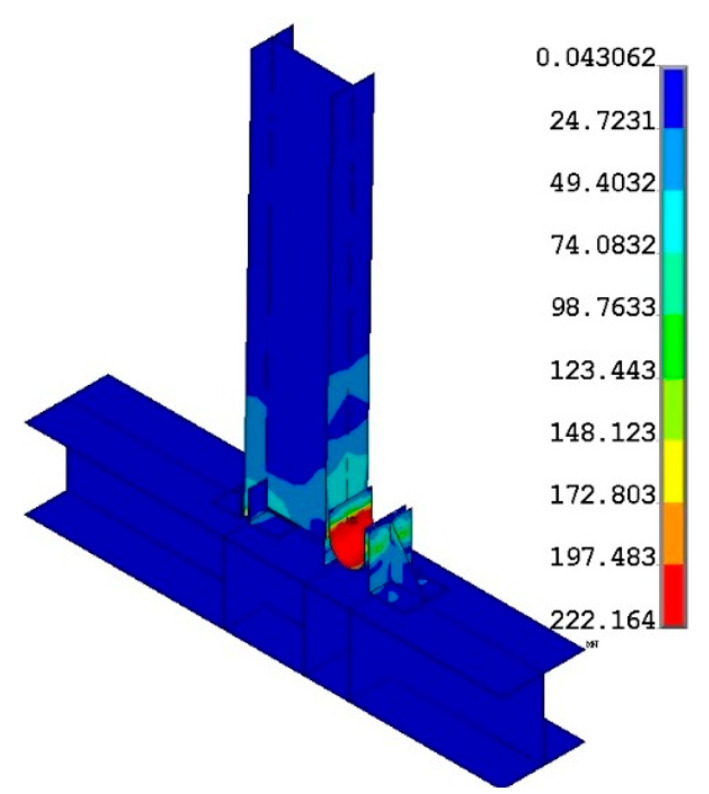
Stress nephogram.

**Figure 15 materials-15-07085-f015:**
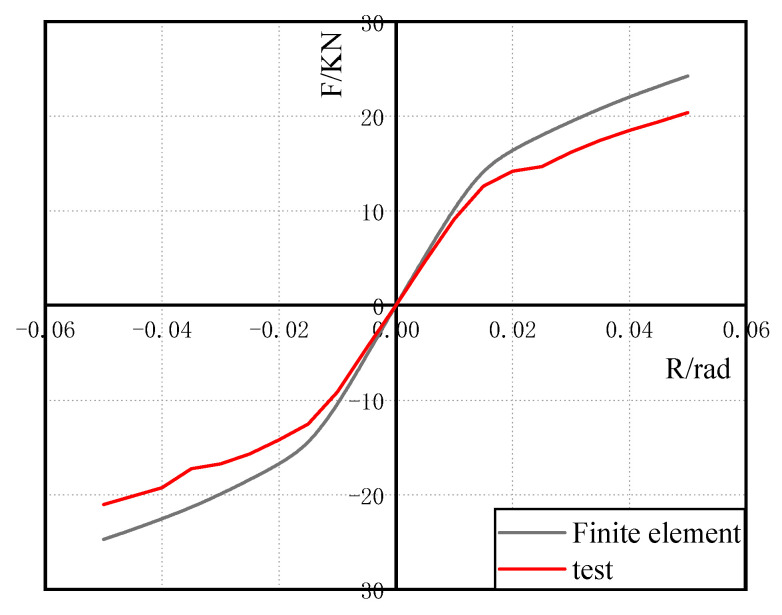
Skeleton curve comparison.

**Figure 16 materials-15-07085-f016:**
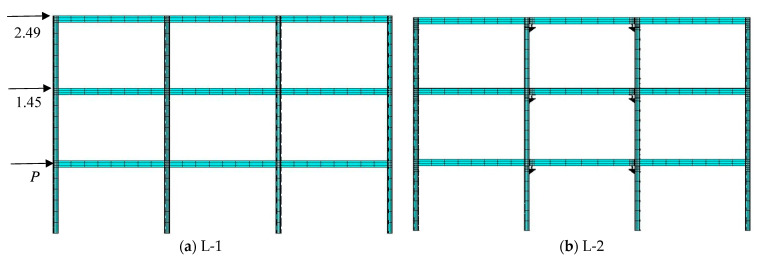
A multi-story rigid frame finite element model.

**Figure 17 materials-15-07085-f017:**
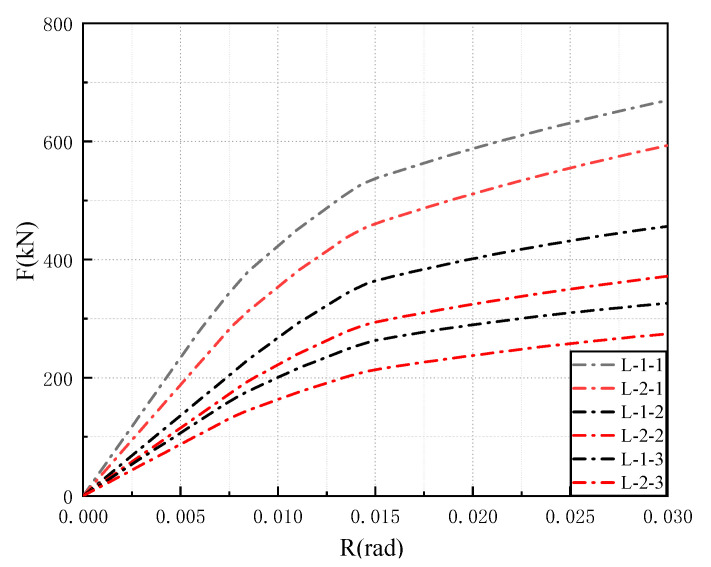
The load-displacement curves of model L-1 and L-2.

**Figure 18 materials-15-07085-f018:**
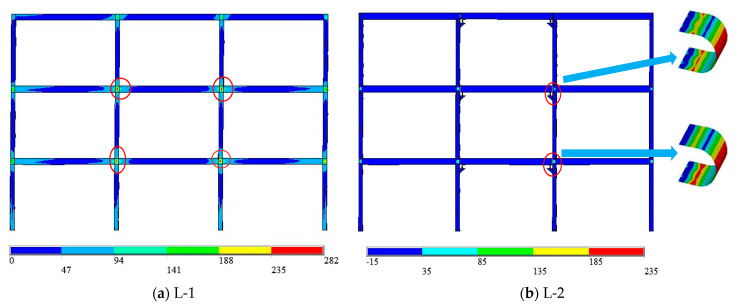
Stress nephogram of multi-story frame.

**Table 1 materials-15-07085-t001:** Construction parameters.

Size	U-Shaped Steel Damper				Beam			Column
	r (mm)	w (mm)	w (mm)	l (mm)	H (mm)	$d_{ab}$ (mm)	$d_{bc}$ (mm)	$l_{b}$ (mm)
	60	175	20	80	1500	250	220	1800

**Table 2 materials-15-07085-t002:** Material properties.

	Materials	Yield Strength (N/mm^2^)	Ultimate Strength (N/mm^2^)	Elastic Modulus/MPa	Ultimate Strain
Beam, Column, Angle steel	Q345B	384.31	543.28	209000	0.04
U-shaped steel damper	Q235B	306	471	205460	0.04
High-strength bolt	10.9S	940	1040	212000	0.05

**Table 3 materials-15-07085-t003:** Inaccuracy analysis.

Initial Stiffness (kN/rad)			Yield Load (kN)		
Theory	Test	Inaccuracy (%)	Theory	Test	Inaccuracy (%)
969.56	1012	4.45	11.10	13.11	18.21

**Table 4 materials-15-07085-t004:** Equivalent viscous damping coefficient.

Interlaminar deformation angle (rad)	0.01	0.015	0.02	0.03
Equivalent viscous damping coefficient	0.0561	0.836	0.1461	0.2877

**Table 5 materials-15-07085-t005:** Ductility coefficient.

Parameters	*X* _u_	*X* _y_	*μ*
Specimens	72	27	2.6

**Table 6 materials-15-07085-t006:** Push-over analysis results.

Model	Story	Connect	Initial Stiffness (kN/rad)	Interlaminar Deformation Angle (rad)	Yield Load (kn)
L-1	1	Rigidity	46824	0.0078	354.26
	2		27200	0.0092	250.92
	3		21328	0.0074	157.83
L-2	1	Rigidity semi-rigidity composite	32682	0.0078	292.08
	2		23019	0.0091	207.17
	3		17441	0.0076	132.55

## Data Availability

Not applicable.

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
