# Peer review of "Mechanical Performance Study of Beam–Column Connection with U-Shaped Steel Damper"

_materials, 2022, doi:10.3390/ma15207085_

Round 1

Reviewer 1 Report

This paper should be major revised. Some comments are as follows:

- In the Table 2: the material of high-strength bolts should be presented; the unit of yield strength and ultimate strength should be replaced by N/mm2.

- In section 4.1. Finite element model:

+ More explanation about modeling should be presented (boundaries, loading, and contact conditions, etc.)

+ For high-strength bolts, which material model was used? It has not mentioned yet.

+ How to choose the suitable mesh size of beam and column? From figure 12, It seems that 25 mm is very big mesh size. Mesh size will significantly affect load-displacement curve. Therefore, a sensitivity analysis should be presented to choose the suitable mesh size.

+ Residual stresses and geometric imperfection of the beam and columns should be considered.

- In Figure 14, there is a big difference between the FE model and test results. The reason may come from some issues: mesh size, residual stresses and geometric imperfection, material properties, etc. The author should carefully check the modeling procedure.

- Failure mode obtained from the FE model and test should be compared.

- It is more interesting if the author verifies the accuracy of the FE model in case of hysteretic curve instead of skeleton curve.

- In section 4.3.2, how to demonstrate the accuracy of the frame structure model?

- Load-displacement curves should be presented for the model L-1 and L-2. This is a good comparison for demonstrating the effectiveness of U-shaped metal damper as well.

- English grammar and typo should be checked.

- Some up-to-date references should be mentioned in the Introduction section.

Reviewer 2 Report

Dear Authors,

Thank you for the opportunity to participate in the review of this article.

The evaluation of this new U-profile damper design for increasing the seismic resistance of steel connections is interestingly prepared and provides good results. 

You state in the article that this is a new type, but several references of a very similar nature can be found in the literature:

Analysis of the Mechanical Properties and Parameter Sensitivity of a U-Shaped Steel Damper July 2021 Frontiers in Materials 8:713221 Follow journal

DOI: 10.3389/fmats.2021.713221

Testing of seismic dampers with replaceable U-shaped steel plates

Bing Qu, Chunxue Dai et al.

Engineering Structures, 179, 1 2019

Etc.

Can you explain the background and differences?

Here are further recommendations:

The title and abstract of the paper are well done - I have minor reservations about the sentence " On the other hand, the correctness of the model is validated by constructing a finite element model at the same scale.", as the meaning is rather contradictory to the previous statement about the correctness of all results - rethink the expression "on the other hand".

At the end of the introductory section you do state what the paper brings, but the justification is not sufficient because in your opinion it is a traditional problem, only there are already more solutions and therefore the question is how your solution is better. 

The geometric description of the damper is understandable. 

Why do you use the terms pseudo-static and quasi-static? choose the appropriate one and use only one. 

The pictures in chapter 3 are small, you need to make them bigger. 

Chapters 3.5 and 3.6 should be a separate supersection as they are about test results and their evaluation.

In Chapter 4 it is not clear what software you used - is it commercial software or your own FEM analysis software?

The model description and evaluation should better describe the situation itself - boundary conditions, network density, etc. Similar to e.g. in:

https://doi.org/10.3390/ma14216573

The variance of 20% for comparing analytical and experimental results is quite large. 

The conclusions are a bit weak given the results you provide. 

Reviewer 3 Report

This research presented an experimental and numerical investigation of a steel beam-column connection using a U-shaped steel damper under a cyclic loading scenario (seismic loading). Generally, the paper has well discussed the case of study and the outcomes. The authors need to consider the following comments for more improve the paper’s presentation:

1. In order to clearly discuss the objective of the suggested research, then the main gap and/or difference between this study with the existing studies needs to be highlighted clearly in the Abstract. 

2. The abstract needs to present the key results from the experimental and numerical results.

3. The introduction is well discussed the background research. However, the problem statement and the objectives of the suggested research need to be highlighted and clearly discussed extensively.

4. Pages 4 and 5: The titles of Figures 3 and 4 need to be provided under the related picture/sketch. The pictures are presented not in line with the related titles.

5. Section 4; provide the name of finite element software, types of surface interaction between the steel parts, and materials constitutive models. The section related to the proposed FE models must be improved to give more details related to the modeling work.

6. The contribution of the suggested FE modeling work needs to be highlighted clearly in the paper.

7. In the conclusion section highlight the main key outcomes (results) that were obtained from the experimental and numerical investigations.

8. In general, the limitation of the current study needs to be clearly mentioned and discussed in the introduction and/or in the conclusions.

Round 2

Reviewer 1 Report

Some comments have not clarified as follows:

Point 2: Loading condition has not mentioned. The way to apply load should be clearly presented.

Point 3: Material constitutive model of the high-strength bolts has not presented. Figure 12 was used for all steel members only or it was also used for the high-strength bolts? It has been noted that the material model for the high-strength bolts is different from that for normal steel members. 

Point 4: Sensitivity analysis should be presented in the revised manuscript.

Point 5: The work that was presented in the manuscript related to nonlinear inelastic behavior not elastic stage as the author mentioned since in the modeling, the author considered geometric and material nonlinearities. The material model used in figure 12 means that material nonlinearity was considered.

Point 6: In figure 15 of the revise manuscript, the curve included elastic and plastic stages, the author should clearly clarify the behavior of the connection considering nonlinear inelastic behavior. The different of the test and simulation may come from some reasons that the reviewer mentioned. 

Point 7: If nonlinear inelastic behavior was considered. The failure model obtained from the test and simulation should be presented in the comparison to demonstrate the accuracy of the proposed model.

Point 8: If the author only considered the elastic stage, the material model used in Figure 12 could not be used. This material model was used to consider the plastic stage of the model.

Point 9: Since the frame was modeled considering nonlinear inelastic behavior (presented in Figure 17), it is hard to demonstrate the accuracy of this frame model without experiment testing. The behavior of the frame model is totally different from the connection verified.

Reviewer 2 Report

You could be inspired by some of these articles or others:

10.3390/su12145691

10.3390/buildings12040433

Even without this comment, the article is suitable for publication. 

Reviewer 3 Report

The authors adequately addressed the given comments, thus the paper in the updated version found accepted 
